# Need and Importance of Nutrition Informatics in India: A Perspective

**DOI:** 10.3390/nu13061836

**Published:** 2021-05-27

**Authors:** Ashish Joshi, Ann Gaba, Shyamli Thakur, Ashoo Grover

**Affiliations:** 1Population Health Informatics, City University of New York Graduate School of Public Health and Health Policy, New York, NY 10027, USA; 2Department of Environmental, Occupational and Geospatial Health Sciences, City University of New York Graduate School of Public Health and Health Policy, New York, NY 10027, USA; ann.gaba@sph.cuny.edu; 3Foundations of Healthcare Technologies Society, New Delhi 110066, India; shyamli108@gmail.com; 4Indian Council of Medical Research, New Delhi 110029, India; ashoogrover@gmail.com

**Keywords:** nutrition informatics, digital health, workforce, food and nutrition

## Abstract

Nutrition informatics (NI) is the effective retrieval, organization, storage, and optimum use of information, data and knowledge for food-and-nutrition-related problem-solving and decision-making. There is a growing opportunity to facilitate technology-enabled behavioral change interventions to support NI research and practice. This paper highlights the changing landscape of food and nutrition practices in India to prepare a NI workforce that could provide some valuable tools to address the double burden of nutrition. Management and interpretation of data could help clarify the relationships and interrelationships of diet and disease in India on both national and regional levels. Individuals with expertise in food and nutrition may receive training in informatics to develop national informatics systems. NI professionals develop tools and techniques, manage various projects and conduct informatics research. These professionals should be well prepared to work in technological settings and communicate data and information effectively. Opportunities for training in NI are very limited in developing countries. Given the current progress in developing platforms and informatics infrastructure, India could serve as an example to other countries to promote NI to support achieving SDGs and other public health initiatives.

## 1. Introduction

### 1.1. Brief History of Nutrition Informatics

At the beginning of the world wide web (WWW), some of the most popular websites were about food and nutrition information. These were operated by government agencies, educational institutions and professional societies, or the food industry and related commercial interests. These two types of websites pointed out the potential for sharing information via the WWW and called upon the newly coined practice of “nutrition informatics” (NI) to help the public parse this new deluge of information [1]. Other authors echoed this sentiment and called upon health professionals to apply informatics to managing nutrition information available on the world wide web [2]. It was indicated that studying informatics will become as fundamental to the practice of medicine and other related disciplines as anatomy has been to the last century [2]. The American Dietetic Association (now the Academy of Nutrition and Dietetics) initiated their work on NI shortly thereafter. The first discussion of NI applications was published in 2006 [3]. They defined NI as a specialty in the field of human nutrition and dietetics that integrates science, evidence-based practice, research, computer knowledge, and expertise in electronic information systems for the purpose of supporting optimal nutritional status and health. The definition of NI expanded as an effective retrieval, organization, storage, and optimum use of information, data, and knowledge for food and nutrition-related problem-solving and decision-making [4]. Informatics is supported by using information standards, information processes, and information technology [4]. NI is connected to the nutrition care processes and using standardized language for documentation of nutrition care. The development of standardized terminology facilitates data retrieval, which, in turn, facilitates billing for services and research to evaluate outcomes and improve patient care [5]. NI can become the tool to manifest problem-solving in these areas.

To determine the competencies necessary for the practice of NI, a Delphi study across all levels of nutrition and dietetics practice was conducted in 2012 [6]. The groundwork for education and training in this area is laid out to differentiate the skills expected for practitioners from entry-level through expert. Several surveys followed developing NI among Academy members [7,8]. The Commission on Dietetic Registration (CDR) in 2016 included informatics as an essential practice competency in continuing education for Registered Dietitian Nutritionists [9]. Most recently, the Accreditation Council on Education in Nutrition and Dietetics (ACEND) included CRDN 4.4 “Apply current nutrition informatics to develop, store, retrieve and disseminate information and data to the Accreditation Standards for Nutrition and Dietetics Internship Programs for 2017 [10]. NI included in the standards for the future education model at the bachelor’s and master’s degree levels [11,12]. The Academy released a formal position statement on NI in 2019 [13]. In addition, the American Society for Parenteral and Enteral Nutrition (ASPEN) collaborated with the Academy to produce a web-based NI resources toolkit [14].

### 1.2. Impact of Legislation in the United States

The widespread adoption of electronic medical records (EMRs) in clinical practice settings was one of the primary drivers for developing NI. The Health Insurance Portability and Accountability Act (HIPAA) of 1996 underscored the importance of maintaining the privacy and security of patient information in medical records [15]. The need to document and retrieve information from medical records in ways that were consistent with HIPAA requirements spurred developing NI practices by the Academy of Nutrition and Dietetics [13,16]. The Health information technology for Economic and Clinical Health (HITECH) act of 2009 provided further impetus for developing NI. This legislation is designed to incentivize the growth of health information technology and promote the adoption and meaningful use (MU) of health data [17]. One component of the HITECH Act was the specific goal of utilizing health information as a tool for continuous improvement of healthcare services. This feature was of great importance to dietitians and developing NI [18].

### 1.3. Global Nutrition Informatics Applications

To make documentation of clinical nutrition care more compatible with NI initiatives, the Academy of Nutrition and Dietetics formulated the International Dietetics and Nutrition Terminology (IDNT) [18]. The goals of having and using standardized language are to promote clear communication across facilities, and to identify in a consistent manner the nutrition problems observed, interventions employed, and outcomes. This facilitates the ultimate goal of NI, which is to improve health outcomes [19]. Data collected through NI methods are anticipated to be a means to provide evidence-based nutrition guidance in various settings [20,21,22,23]. These include clinical practice [24], coding and billing for nutrition care services [25], food service management [26], food composition research [27], and food and nutrition policy [28]. The Academy of Nutrition and Dietetics Health Informatics Infrastructure (ANDHII) was created [29] and pilot tested [30] to begin establishing a research framework to support informatics analysis of nutrition intervention and outcomes data.

The utilization of NI has been presented as a global challenge [31]. The Dietitians Association of Australia has begun implementing NI activities similar to those in the United States [32,33]. This has also been encouraged by the European Federation of Associations of Dietitians, with dietitians in Switzerland [34], Sweden [35] and Norway [36] implementing these practices. Work has been conducted for using standardized methods and terminology and translating these into multiple languages for use in countries, such as Canada, Denmark, Greece, Ireland and New Zealand [37]. Population health informatics in other contexts has been applied to the exploration of nutrition issues worldwide. For example, assessment of nutrition in children as part of population health surveillance in Ethiopia [38] and monitoring of population rates of nutrition-related non-communicable diseases in Peru [39].

### 1.4. Food and Nutrition Landscape in India

India is undergoing a nutrition transition, driven both by demographic and socioeconomic changes [40]. Understanding the causes and consequences of these changes could be important in preventing diet-related diseases of both under- and over-nutrition [41]. Assessment of food consumption patterns around the country found an overall increase in processed foods and snack foods, with increases in consumption of sweets varied by region [42]. Changes from traditional dietary patterns have been associated with increasing the incidence of obesity and diabetes throughout the country, especially in urban areas [43,44]. The development of appropriate nutrition guidance remains challenging. One reason for this is the diversity of dietary patterns and cooking methods regionally throughout the country, making identifying an “average” Indian diet virtually impossible [45]. An analysis of dietary patterns found twenty-nine distinct diet patterns, each one differently related to the range of nutrition problems experienced in India [45].

This variability has made implementing nutrition interventions to prevent diseases, such as diabetes, more challenging [46]. Furthermore, the rapidly changing food and agriculture environment makes estimating dietary intake a constantly moving target [47], with changes to agriculture to promote health and environmental impacts [48]. Despite these many challenges, standardized terminology and NI have not yet been adopted by the Indian Dietetic Association [49].

## 2. Discussion

### 2.1. Need for Nutrition Informatics in India

Given the volume and complexity of the information and the urgency of developing solutions to the health problems encountered within a transitioning food environment, NI could provide some valuable tools [50]. Management and interpretation of big data could help to clarify the relationships and interrelationships of diet and disease as these relate to the socioeconomic and agricultural changes occurring in India on both national and regional levels. Data mining could yield targeted, evidence-based recommendations and guide interventions.

The US National Academy of Sciences has laid out a National Food Data System [51]. They state that features of this system should include comprehensiveness, representativeness, timeliness, openness, flexibility, accuracy, suitability for causal analysis, and fiscal responsibility. The goals of this system would be to have a reliable, multisource, interconnected data system to support research and to guide policy. Data from both national and local government sources, along with those of food industry sources, can be coordinated into a similar structure for India. NI data can promote understanding of the drivers of food choices in various regions, how those choices affect health outcomes, and the likely efficacy of implementing food and nutrition policies.

The descriptive information collected in this review can be viewed through the lens of a strengths, weaknesses, opportunities and threats (SWOT) analysis that is commonly applied by businesses. These are summarized in Table 1 below.

### 2.2. Digital India Initiative

The Digital India Initiative’s goal is to create infrastructure as a utility through which every citizen can access multiple public services through a single integrated window. Several cloud platforms using open Application Programming Interfaces (APIs), such as Seva, PayGov, eSangam, and MeghRaj established to provide software interoperability for all e-governance systems and applications. Geospatial information systems (GIS) enhance delivering government services through the cloud or mobile technology. The interoperability of these applications will support both new initiatives and the centralization of access to existing programs and agencies [52].

### 2.3. Nutrition Informatics Platforms in India

Multiple platforms currently exist or are under development to collect, tabulate and report nutrition information in India.The PM’s Overarching Scheme for Holistic Nourishment (*POSHAN*) *Abhiyan* is a comprehensive platform launched in 2018 by the National Nutrition Mission. It aims to improve nutritional outcomes for children, pregnant women and lactating mothers. Its purpose is to address malnutrition, undernutrition, anemia, low birth weight babies and stunting of children’s growth through coordination of services through the first 1000 days of life, from conception to the child’s second birthday. One of the key aims of POSHAN Abhiyan is to promote coordination of nutrition action plans at the state, district, and block levels. Convergence action plan (CAP) committees have been created to plan, coordinate, and operationalize services across multiple departments and to assess outcomes and impacts in the community [53].Integrated Child Development Services—Common application software (CDS-CAS) introduced by POSHAN Abhiyan is a web and mobile phone-based application to improve nutrition outcomes, service delivery and program management. The application enables Anganwadi workers (AWWs) to track service delivery and make informed decisions by facilitating AWWs in their job tasks and responsibilities, helps supervisors to assess and offer feedback to them. The ICDS-CAS has three components—a mobile-based application for AWWs, one for supervisors, and a web-based dashboard for program officials. It auto plots the growth chart on the mobile application, enabling growth monitoring of children, auto-generates a task list, and is useful in counseling during home visits. The home visit scheduler enables AWWs to focus on the clients. Growth monitoring devices (GMDs) ensure accurate records of weight and height. Children six months to six years of age are weighed, and their height is recorded every month to monitor their growth trend, following which SMS alerts are sent to parents of children recording static growth. AWWs reported challenges in using the application largely related to the application, hardware or network issues [54,55]. POSHAN Abhiyan has developed educational activities for Anganwadi workers (AWW) and their supervisors through the training and capacity-building program. For this training, workers are provided with smartphones loaded with monitoring software available in fifteen languages. Training is conducted through twenty-one thematic modules, available through a web-based portal and utilizing an incremental learning approach (ICA) [56]. Furthermore, assessing program outcomes requires accurate and consistent data on food consumption and micronutrient content of foods consumed. All related data tools and aids for this task must be available through a single easily accessible platform [57].Anemia Mukt Bharat Dashboard is an example of data collection through POSHAN Abhiyan programs. This initiative collects data on six interventions, all of which focus on the prevention of anemia. These programs include the provision of iron and folic acid supplements to children, adolescents, women of child-bearing age, and pregnant and lactating women; the annual National Deworming Day program for children one to nineteen years of age; a year-round behavior change communication campaign, targeting practices related to anemia prevention; anemia testing and treatment for pregnant women and school-going adolescents; provision of iron and folic acid fortified foods in government-funded health programs; and screening and treatment of non-nutritional causes of anemia with a focus on malaria, hemoglobinopathies and fluorosis [58].Jan Andolan Dashboard is another example of a mobile application that reports on community-based events, social and behavioral change events, and district-planned activities [59].Data on the nutritional status of children in India are often only available at the national and state levels, with very little data present at the district level. POSHAN has generated district nutrition profiles (DNPs) for the 640 districts of India to facilitate evidence-based discussions about undernutrition. DNPs provide a snapshot of the state of nutrition in each district [60]. Besides POSHAN Abhiyan data, several other nutrition platforms established to provide and monitor nutrition analytics.The Food Safety and Standards Authority of India (FSSAI) is charged with ensuring safe, healthy, sustainable and wholesome food for the people in India. To this end, the agency has initiated many efforts under the Eat Right India program. The food safety measures, such as establishing a hygiene-rating system for restaurants and catering establishments and sustainability initiatives aimed at promoting environment-friendly food production practices and building consumer awareness of healthy food choices. Programs like Eat Right Campus and Eat Right School were initiated to target individuals in workplaces, colleges, universities, institutes, hospitals, tea estates and jails, as well as children in schools. The Eat Right Toolkit has also been launched and is disseminated by trained health workers [61].The National Institute of Nutrition (NIN)’s Nutrition Atlas utilizes data from sources like the National Family Health Survey, the National Nutrition Monitoring Bureau, and the World Health Organization (WHO). NIN’s Nutrition Atlas provides information and data on the population’s nutritional status at state and national levels. It gives an overview of nutrition-related deficiencies, disorders and prevalence rates in various parts of the country. It also provides information on nutrients, nutrient-rich foods, nutritional deficiency and disorders. This dashboard also provides temporal trends on undernutrition, overweight, obesity and communicable and non-communicable diseases [62].Nutrify India Now is a mobile app developed by the NIN, which guides in assessing the nutrients provided by foods consumed. It contains comprehensive nutritional information on Indian foods and on common Indian recipes to track total calories, protein, vitamins and minerals. The app enables users to look up foods and ingredients rich in any specific nutrient of interest. The database includes the names of raw foods in 17 Indian languages and uses Indian guidelines set forth by the Indian Council of Medical Research. This app is available as a free download in online app stores [63].Tata Trusts, a private international holding company with corporate headquarters in Mumbai, collaborated with NIN to establish Tata-NIN Center for Excellence in Nutrition. This center aims to create a database on the production, distribution and intake of foods in India and their relationship to disorders of nutritional excess or deficiency. The center will also support research on nutrition and health across multiple domains [64].Nutrition India is the Ministry of Health and Family Welfare’s dashboard on the child, adolescent and maternal nutrition. It compiles nutrition information from national surveys and the Health Management Information System (HMIS). This tool provides maps, tables and trends at the district, state and national levels [65]. One such platform is the Champions of Change Dashboard, which includes ranked health and nutrition data in nutritional and state data sets [66].The National Institute for Transforming India (NITI) Aayog State Nutrition Dashboard reports state-level information on various child nutrition parameters and compares it to ten-year averages based on the National Family Health Survey (NFHS) [67].Nutrition on My Radar Screen is an interactive data visualization tool that provides visualizations of data on forty-four sub-indicators used by NFHS-4 for domains of maternal and child health and nutrition. Based on the specific categorization of data points, states ranked as having good, neutral, or bad performance on a given variable [68,69].

### 2.4. Nutrition Informatics Workforce Needs in India

Nutrition has been identified as having a fundamental role in sustainable development. The United Nations has developed Sustainable Development Goals (SDGs) [70]. Twelve of the seventeen SDGs contain indicators that are pertinent to nutrition. Of these, goal two aims to end hunger, achieve food security, improve nutrition, and promote sustainable agriculture. Progress towards meeting the SDGs requires ongoing evaluation of nutritional status indicators. Strong data collection methods must be employed to address nutrition issues such as stunting, wasting, and obesity [71]. Collection analysis and reporting of these require an understanding of nutrition informatics. To compare national data to international benchmarks, it is important to collect and report high-quality data. In a message for the Scaling Up Nutrition (SUN) Movement Strategy and Roadmap 2016–2020, UN Secretary-General Ban Ki-Moon said, “*Improved nutrition provides the platform for progress in health, education, employment, empowerment of women, and reduction of poverty and inequality for which sustained and adequate investments in good nutrition is the prerequisite for achieving SDGs*” [72].

According to the Global Nutrition Report, worldwide data gaps are a major hindrance to the betterment of nutrition. Without accurate data collection and assessment, the impact of expenditures to counteract stunting, wasting, anemia, obesity and non-communicable diseases remains unknown. This requires coordinated and convergent efforts by all related ministries, departments and stakeholders [71,73]. Although various such efforts have been implemented in India, there remains a need for this type of coordination of efforts at a national level. Without proper identification of a workforce, their jobs and responsibilities, the capacity needs for an efficient process and outcome evaluation are unmet, and nutrition policies fail to deliver the desired health and nutrition outcomes [74]. Collection and coordination of data need alignment with goals and objectives consistent with the SDGs. In turn, developing numerical and time-bound targets must be aligned with the collection of appropriate data on all levels [71]. The creation of such a system requires a skilled workforce to optimize these undertakings.

NI professionals and specialists develop tools and techniques, manage various projects and conduct informatics research to analyze the efficacy of the tools and processes developed [6]. Individuals with expertise in food and nutrition may receive training in informatics with a skill level sufficient to provide the leadership needed to develop national informatics systems [5]. These professionals should be well prepared to work in technological settings and be capable of utilizing, documenting and communicating data and information effectively [6]. NI is an emerging area of specialization for food and nutrition professionals. Programs in NI establish processes to ensure incorporating suitable technologies and information systems and educating dietetic professionals [32]. Integration of informatics systems into nutritional care enhances the quality of nutrition care and efficiency of the professionals [75]. The application of informatics improves the efficiency and quality of work of dietitians in meal planning and menu assessment [76]. Advancing NI beyond a basic level will require integrating skills in selection, utilization and management of suitable technology [6] as well as skills in guiding consumers in applying the information available to them through informatics systems [77]. To provide this level of expertise, education and training in NI have been proposed across various educational levels, including undergraduate [78], graduate [6] and professional continuing education [77]. Proposed curricula include content on standardized terminologies, systems design, standards and regulations related to technology, as well as applications, including analysis, management, and evaluation [6].

Limited training opportunities are available in these critical areas of NI in developing countries, despite the demonstrated need for expertise in this area to address the SDGs. Virtual education can foster the needed skill development in the workforce. Given the current progress in developing platforms and informatics infrastructure, India could serve as an example to other countries to promote NI to support achieving the SDGs and other public health initiatives. To realize this, a workforce skilled in NI is key.

Digital health training initiatives should focus on competencies relevant to a particular healthcare worker group, role, level of seniority, and setting. Several reasons exist to train and educate healthcare workers to be digitally competent. First, the growing use of digital technology in healthcare intensifies the need for capacity building and continuous professional development. Second, the importance and potential of remote care were brought to light recently with the COVID-19 pandemic [79]. Virtual consultation devices and electronic systems are indispensable tools to diagnose and treat patients with potential COVID-19 infections and all other infections [80,81]. Third, surveys of healthcare workers show that they would appreciate more training on digital technology [82,83,84]. Improving digital literacy capabilities could lead to better digital services and technologies in healthcare settings [85].

Although the value of NI is clear, competing priorities may limit the ability of relevant agencies to implement all of the features that have been identified as being important to the growth of NI in India. It is important to recognize competing priorities where they exist and to promote applying NI strategies concurrently wherever possible with approaches to other social and public health issues.

## 3. Conclusions

Digital health technologies can be key to improving health outcomes, provided healthcare workers are adequately trained to use them. Future training programs need to consider the evolving nature of digital health and incorporate upcoming digital trends. Continued efforts to increase the awareness of nutrition informatics and HIT benefits among dietitians are crucial.

## Figures and Tables

**Table 1 nutrients-13-01836-t001:** SWOT analysis.

Strengths	Opportunities
Existing infrastructure and programs	Needs that can be addressed by NI
Digital India Initiative	NI can help to clarify the relationships and interrelationships of diet and disease as these relate to the socio-economic and agricultural changes occurring in India on both national and regional levels.
POSHAN Abhiyan	
Eat Right India program - Eat Right Campus - Eat Right School - Eat Right Toolkit	Data from both national and local government sources, along with that of food industry sources can promote understanding of the drivers of food choices in various regions, how those choices affect health outcomes, and the likely efficacy of the implementation of food and nutrition policies.
National Institute of Nutrition - Nutrition Atlas - Nutrify India Now mobile app	
Tata-NIN Centre for Excellence in Nutrition	Data mining with NI could yield targeted evidence-based recommendations and guide interventions.
Nutrition India dashboard on child, adolescent and maternal nutrition	
The National Institute for Transforming India Aayog State Nutrition Dashboard	India could serve as an example to other countries as to how to promote NI to support achievement of the SDGs and other public health initiatives.
Nutrition on my Radar Screen interactive data visualization tool	
**Weaknesses**	**Threats**
Limitations in workforce and training opportunities and gaps in existing data	Competing priorities in India
The NI workforce remains undefined, and opportunities for training are very limited in developing countries.	The COVID 19 pandemic
Standardized terminology and NI have not yet been adopted by the Indian Dietetic Association.	Responses to natural disasters
The diversity of dietary patterns and cooking methods regionally throughout the country making identification of an “average” Indian diet challenging.	Local and regional competition for resources with varied priorities other than NI
Gaps in data limit accurate data collection and assessment which guide expenditures to counteract stunting, wasting, anemia, obesity and non-communicable diseases.	

## Data Availability

Not applicable.

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
