# Peer review of "Need and Importance of Nutrition Informatics in India: A Perspective"

_nutrients, 2021, doi:10.3390/nu13061836_

Round 1
Reviewer 1 Report
Dear Authors,
The manuscript (nutrients-1200645) presented for review is quite interesting. I don’t sure but probably not for every reader. Authors only inform about Nutrition Informatics in India. It will be more interesting if the authors try to do a SWOT analysis of the different types of Nutrition Informatics in India, and pointed the direction for India in this topic and possible solutions.
I recommend the article for publication but after major revision.
In the whole manuscript, there is a lack of chapter numbers.
References
Authors, Please check the correctness of the citation of references by requirements of the Nutrients Journal. In my opinion, now it is not correct. For example, 55 and 56 are probably the same reference, 59 and 60 are the same situation, but without information about the date of access.
The font of the reference number is not well written.
Despite my comments, I recommend this manuscript for publication but after depth revision, because this form is not acceptable. I believe it addresses an important area of research in an international context and proper nutrition in India.
Best of luck with your paper and be safe!
Reviewer
Author Response
The authors would like to thank both reviewers for their time in reading this manuscript, and their thoughtful comments pertaining to our work.

Reviewer 2 Report
Dear Authors -
Thank you for giving me an opportunity to review your communication brief. Good work on this effort. My comments are provided in the PDF. Good luck!

Author Response

(The authors gave the same response as above.)

Round 2
Reviewer 1 Report
Dear Authors,
The authors have changed sections Introduction and Discussion of the planned paper in line with my and other reviewer suggestions. Moreover, the authors corrected English and many details in the manuscript. In my opinion, now the manuscript is clear and better understand than the previous version.
I would like to thank the authors for considering my comments and applaud them for the major revisions to improve their manuscript. Especially thank you for the interesting SWOT analysis.
Now, I have the pleasure to recommend the manuscript for publication without correction.
Reviewer